# Spinel-Type (FeCoCrMnZn)_3_O_4_ High-Entropy Oxide: Facile Preparation and Supercapacitor Performance

**DOI:** 10.3390/ma13245798

**Published:** 2020-12-18

**Authors:** Bingliang Liang, Yunlong Ai, Yiliang Wang, Changhong Liu, Sheng Ouyang, Meijiao Liu

**Affiliations:** 1Key Laboratory for Microstructural Control of Metallic Materials of Jiangxi Province, Nanchang Hangkong University, Nanchang 330063, China; lbl@nchu.edu.cn (B.L.); 1801085204040@stu.nchu.edu.cn (Y.W.); 27014@nchu.edu.cn (C.L.); 2School of Materials Science and Engineering, Nanchang Hangkong University, No.696, South Fenhe Avenue, Nanchang 330063, China; 70685@nchu.edu.cn (S.O.); A9819120525@163.com (M.L.)

**Keywords:** (FeCoCrMnZn)_3_O_4_, high-entropy oxide, spinel structure, supercapacitor

## Abstract

High-entropy oxides (HEOs) have attracted more and more attention because of their unique structures and potential applications. In this work, (FeCoCrMnZn)_3_O_4_ HEO powders were synthesized via a facile solid-state reaction route. The confirmation of phase composition, the observation of microstructure, and the analysis of crystal structure, distribution of elements, and valences of elements were conducted by X-ray diffraction (XRD), scanning electron microscopy (SEM), transmission electron microscopy (TEM), energy-dispersive X-ray spectroscopy (EDS), and X-ray photoelectron spectroscopy (XPS), respectively. Furthermore, a (FeCoCrMnZn)_3_O_4_/nickel foam ((FeCoCrMnZn)_3_O_4_/NF) electrode was prepared via a coating method, followed by the investigation of its supercapacitor performance. The results show that, after calcining (FeCoCrMnZn)_3_O_4_ powders at 900 °C for 2 h, a single spinel structure (FCC, Fd-3m, *a* = 0.8399 nm) was obtained with uniform distribution of Fe, Co, Cr, Mn, and Zn elements, the typical characteristic of a high-entropy oxide. In addition, the mass specific capacitance of the (FeCoCrMnZn)_3_O_4_/NF composite electrode was 340.3 F·g^−1^ (with 1 M KOH as the electrolyte and 1 A·g^−1^ current density), which indicates that the (FeCoCrMnZn)_3_O_4_ HEO can be regarded as a prospective candidate for an electrode material in the field of supercapacitor applications.

## 1. Introduction

The concept of “high entropy” is a new material design concept originally developed from high-entropy alloys in recent years, and it has become a hotspot in the field of material research. In 2004, Ye et al. first proposed the concept of high-entropy alloys, in which a variety of alloying elements were removed in an equimolar or near-molar ratio to obtain a solid solution with a single crystal structure [1]. Owing to the difference in alloy elements and the radius of each atom, unexpected effects are produced in highly disordered multicomponent systems: the high-entropy effect, cocktail effect, sluggish diffusion effect, and lattice distortion effect [2,3,4]. The synergy of these four effects leads to alloys exhibiting high strength, high hardness, and excellent corrosion resistance [5,6,7].

With the continuous deepening of research, Rost et al. applied “high entropy” to ceramic materials in 2015 [8]. Recently, this concept was extended to new high-entropy materials such as high-entropy nitride [9], high-entropy carbide [10,11,12], high-entropy boride [13,14,15], and high-entropy sulfide [16], which show unique physical properties and good application prospects in a range of applications.

Initially, Rost et al. prepared a (CoCuMgNiZn)O solid solution with a rock-salt structure (Fm-3m) via a solid-state reaction route [8]. Moreover, they pointed out that the high-entropy oxide (HEO) system has a higher configuration entropy (S) when the metal atoms are in equimolar concentration, resulting in lower Gibbs free energy (G), which indicates that a higher configuration entropy can keep the HEO more stable at high temperature. So far, researchers have synthesized a variety of single-phase HEOs with different crystal structures, such as rock-salt type [8,17,18], spinel type [19,20,21,22], fluorite type [23,24], and perovskite type [25,26,27]. Mao et al. [19,21] prepared spinel-type high-entropy oxides (CrFeMnNiZn)_3_O_4_ and (CrFeMnNiZn)_3_O_4_ via a solution combustion synthesis method with complicated procedures. Up to now, only a small number of HEO-related performances have been reported [18,28,29,30]. Recently, works from Sarkar [18] and Berardan [31] showed that HEOs have great application potential in energy storage. The high configuration entropy can maintain the stability of the original crystal structure of HEOs during charging and discharging such that the cycle stability of HEOs is much higher than that of traditional transition-metal oxides.

In this study, (FeCoCrMnZn)_3_O_4_ HEO powder with a single spinel structure was prepared via a facile solid-state reaction route (900 °C, 2 h). In addition, the supercapacitor performances of a (FeCoCrMnZn)_3_O_4_/Ni foam (NF) electrode were investigated.

## 2. Experiments and Procedures

### 2.1. Synthesis of (FeCoCrMnZn)_3_O_4_ Powders

In this work, a facile solid-phase reaction method was used to synthesize (FeCoCrMnZn)_3_O_4_ HEO powders. First, Fe_2_O_3_ (99.0%), Co_2_O_3_ (99.0%), Cr_2_O_3_ (99.0%), MnO_2_ (91.0%), and ZnO (99.99%) powders (Sinopharm Chemical Reagent Co., Ltd., Shanghai, China), with a molar ratio of 1:1:1:2:2, were mixed using pot mill (GMJ-5, Xianyang Jinhong General Machinery Ltd., Xianyan, China) for 12 h in distilled water, with ZrO_2_ balls as the mill medium. Second, the mixed powders were calcined at 800–1000 °C in the atmosphere (with a heating rate of 10 °C/min) for 2 h, before grinding and sieving after furnace cooling.

### 2.2. Preparation of (FeCoCrMnZn)_3_O_4_/NF Electrode

To prepare the (FeCoCrMnZn)_3_O_4_/NF electrode, as-synthesized (FeCoCrMnZn)_3_O_4_ powders, binder (polytetrafluoroethylene, PTFE), and acetylene black (with a mass ratio of 8:1:1) were mixed in a mixture of ethanol and distilled water via ultrasonic separation for 15 min. Then, the NF was immersed into the as-mixed slurry with ultrasonic treatment to make sure that the NF was covered uniformly by (FeCoCrMnZn)_3_O_4_ powder. Finally, the (FeCoCrMnZn)_3_O_4_/NF electrode was vacuum-dried at 60 °C.

### 2.3. Material Characterization

The phase analysis was conducted using an X-ray diffractometer (XRD; D8 ADVANCE, Bruker-AXES, Karlsruhe, Germany). The particle size distribution was tested using a laser particle size analyzer (BT-9300H). The microstructure of (FeCoCrMnZn)_3_O_4_ powders was observed using a field-emission scanning electron microscope (FE-SEM; Nova Nano SEM 450, FEI, Brno, Czech). The high-resolution transmission electron microscopy images, selected area electron diffraction (SAED) patterns, and energy-dispersive X-ray spectroscopy (EDS) elemental mapping images were obtained using a high-resolution transmission electron microscope (HR-TEM; Talos F200X, FEI, Brno, Czech). The element valence state was determined via X-ray photoelectron spectroscopy (XPS; Axis Ultra DLD, KRATOS, Manchester, UK).

### 2.4. Electrochemical Measurements

The supercapacitor performance (cyclic voltammetry (CV) curves, galvanostatic charge-discharge (GCD) curves and electrochemical impedance spectra (EIS)) was measured by an electrochemical workstation (CHI660E, Shanghai Chenhua Instrument Co., LTD, Shanghai, China) with 1 M KOH electrolyte. (FeCoCrMnZn)_3_O_4_/NF, Ag/AgCl, and Pt foil served as the working electrode, reference electrode, and counter electrode, respectively. The mass specific capacitance (*C_m_*) was calculated by the following equations:(1)$$C_{m}=\frac{\int idu}{2mv\Delta u}$$
(2)$$C_{m}=\frac{it}{m\Delta u}$$
where *C_m_* (F·g^−1^) is the mass specific capacitance, *i* is the discharging current (A), *t* is the discharge time (s), *v* is the scan rate (mV·s^−1^), Δ*u* is the discharge voltage range (V), and *m* is the mass (g) of (FeCoCrMnZn)_3_O_4_ HEO active materials.

## 3. Result and Discussion

### 3.1. X-Ray Diffraction Analysis

Figure 1 shows the XRD patterns of (FeCoCrMnZn)_3_O_4_ powders calcined at 800–1000 °C for 2 h. Figure 1a illustrates that the major diffraction peaks of (FeCoCrMnZn)_3_O_4_ powders calcined at 800 °C could be identified as a spinel structure. However, other minor phases, eskolaite, cobalt manganese oxide, hetaerolite, and Co_3_O_4_-like spinel, were also observed. When the calcining temperature was increased to 900 °C, the diffraction peaks of the minor phases vanished. Notably, all the diffraction peaks of (FeCoCrMnZn)_3_O_4_ powders calcined at 900 °C and 1000 °C could be identified as a spinel structure with the space group of Fd-3m (PDF#34-0140). The diffraction peaks at 2θ = 18.32°, 30.15°, 35.51°, 37.15°, 43.16°, 53.53°, 57.07°, and 62.67° were consistent with the (111), (220), (311), (222), (400), (422), (511), and (440) planes, respectively. In addition, as the calcining temperature increased, the diffraction peaks of (FeCoCrMnZn)_3_O_4_ powders became sharper and narrower, indicating higher crystallinity. The enlarged XRD patterns in the range of 32–38° are presented in Figure 1b. Notably, with the increase in calcining temperature, the diffraction peak corresponding to the (311) crystal plane gradually shifted to a higher scattering angle, suggesting a constriction of the lattice constant [32]. Moreover, it is apparent that the full width at half maximum (FWHM) corresponding to the (311) crystal plane decreased with the increase in calcining temperature, indicating an increase in average grain size. Figure 1c shows the Rietveld refined XRD pattern of (FeCoCrMnZn)_3_O_4_ powders synthesized at 900 °C. The initial structural model was established on the basis of the spinel structure (Fd-3m), whereby all the observed XRD patterns fit well with the calculated data of the spinel structure, and the calculated lattice constant was 0.8399 nm.

### 3.2. Micromorphology and Structure Analysis

Figure 2 demonstrates the SEM image and particle size distribution diagram of the (FeCoCrMnZn)_3_O_4_ powders calcined at 900 °C for 2 h. Irregular particles with some agglomeration can be observed from Figure 2a. It can be seen in Figure 2b that the particle size of the (FeCoCrMnZn)_3_O_4_ powders was mainly between 0.5 μm and 1 μm with a normal distribution, and the average particle size was 0.65 μm. 

In order to better understand the crystal structure of (FeCoCrMnZn)_3_O_4_ powder, it was further determined by TEM (Figure 3). Figure 3b–g show the energy-dispersive X-ray spectroscopy (EDS) analysis of O, Fe, Co, Cr, Mn, and Zn elements. We can find that all metal elements in the grain were uniformly dispersed, without obvious reunion. All results indicate that (FeCoCrMnZn)_3_O_4_ powder was chemically and structurally homogeneous. In addition, the HR-TEM image of (FeCoCrMnZn)_3_O_4_ powder is shown in Figure 3h. The interplanar space value (0.294 nm) obtained from the HR-TEM images is very close to that of the (220) crystal plane of the spinel structure (PDF#34-0140, 0.296 nm), which matches with the results of XRD analysis (Figure 1a). The fast Fourier transfer (FFT) generated in the same area shows that the powder possessed a typical face-centered cubic (fcc) structure, and the diffraction spots are also marked in Figure 3i. The diffraction pattern could be identified as a face-centered cubic spinel structure, which is in accordance with the XRD pattern. 

Figure 4 is the XPS peak spectrum of each element in the (FeCoCrMnZn)_3_O_4_ powder. The peak spectrum of O 1*s* in Figure 4a shows that the peaks at 530, 531.5, and 532.8 eV corresponded to metal oxygen bonds, surface-adsorbed oxyhydroxide, and surface physical/chemically adsorbed H_2_O, respectively. Figure 4b–f show the XPS spectra of Fe, Co, Cr, Mn, and Zn metal elements. The XPS analysis indicates that Fe, Co, and Mn cations in (FeCoCrMnZn)_3_O_4_ HEO were at the valences of both +2 and +3 states, while Cr was in the +3 state and Zn was in the +2 state.

### 3.3. Electrochemical Performance

Due to the unique structural characteristics of the HEO, it has special significance as a supercapacitor. Cyclic voltammetry (CV) measurements were conducted in a three-electrode cell to investigate the pseudocapacitive performance of (FeCoCrMnZn)_3_O_4_/NF electrode. Figure 5a shows CV curves of the (FeCoCrMnZn)_3_O_4_/NF electrode in the potential range of 0.15–0.5 V at a scan rate from 5 to 100 mV·s^−1^. Apparently, there was a clear symmetric redox peak in the CV curve due to the conversion of MO and MO–OH (M represents the metal elements Co, Ni) during the charge–discharge process [33]. In addition, the shape of the CV curves did not change significantly with the increasing scan rate, which manifests that the structure of the (FeCoCrMnZn)_3_O_4_/NF electrode is helpful for the rapid redox reaction [34]. Moreover, as the scan rate increased, the oxidation and reduction peaks of the CV curve moved to high and low potentials, respectively, demonstrating the limitation of charge transfer [35,36]. According to Equation (1), the mass specific capacitance (*C_m_*) of the (FeCoCrMnZn)_3_O_4_/NF electrode at a scan rate of 5 mV·s^−1^ was 352.9 F·g^−1^.

The performance of the (FeCoCrMnZn)_3_O_4_/NF electrode for supercapacitors was studied with chronopotentiometry tests (0.15–0.48 V) at different current densities (Figure 5b). The *C_m_* calculated by Equation (2) at current densities of 1, 2, 5, and 10 A·g^−1^ were 340.3, 287.9, 301.5, and 281.8 F·g^−1^, respectively. The results show that 82.8% remained of the primal *C_m_* even at a high current density of 10 A·g^−1^, which indicates the prominent rate performance of (FeCoCrMnZn)_3_O_4_/NF electrode. Furthermore, the relatively smooth shape of charge–discharge curves indicates that the (FeCoCrMnZn)_3_O_4_/NF electrode is in accordance with the characteristics of pseudocapacitance (nonlinear charge–discharge), which is triggered by the redox reaction of the active material of the electrode during the rapid charge–discharge process. Additionally, the two slow stages of charge/discharge curves in Figure 5b correspond to the redox peaks in CV curves [37]. As shown in Figure 5c, the discharge specific capacitance featured a downward trend with the increase in current density. This is because the reversible redox reaction is a diffusion process. As a result, at a large current density, there was an obvious diminution in the electrochemical utilization of (FeCoCrMnZn)_3_O_4_ HEO powders [38,39]. The cycling performance of the (FeCoCrMnZn)_3_O_4_ powder material supported on NF was tested at a current density of 5 A·g^−^^1^ (Figure 5d). The result shows that about 69% of the initial specific capacitance remained even after 1000 cycles at a current density of 5 A·g^−1^. To explore the characteristics of the (FeCoCrMnZn)_3_O_4_/NF electrode for supercapacitor performance, EIS measurements were conducted at open-circuit potential over the frequency range of 0.01 Hz and 100 kHz for the (FeCoCrMnZn)_3_O_4_/NF electrode. It can be seen from Figure 5e that the slope of the straight line was large, indicating that the (FeCoCrMnZn)_3_O_4_ HEO had good pseudocapacitance performance. Moreover, there were no observable semicircular arcs in the high-frequency region, manifesting that the charge transfer was dominated by diffusion of ions, instead of an interface reaction [40]. Hence, the newly developed (FeCoCrMnZn)_3_O_4_ HEO has outstanding application prospects in supercapacitor applications.

## 4. Conclusions 

In summary, a (FeCoCrMnZn)_3_O_4_ HEO with a single spinel structure was prepared via a facile and easily industrialized route (900 °C, 2 h). The (FeCoCrMnZn)_3_O_4_ HEO exhibited a face-centered cubic crystal structure with five highly dispersed metal elements (Fe, Co, Cr, Mn, and Zn). In addition, the (FeCoCrMnZn)_3_O_4_ HEO can be used as a prospective candidate material for supercapacitors. The mass specific capacitance was 340.3 F·g^−1^ at a current density of 1 A·g^−1^ in the electrolyte of 1 M KOH. This research provides not only a facile route for preparing an HEO, but also a new path for the application of HEOs to the field of supercapacitors.

## Figures and Tables

**Figure 1 materials-13-05798-f001:**
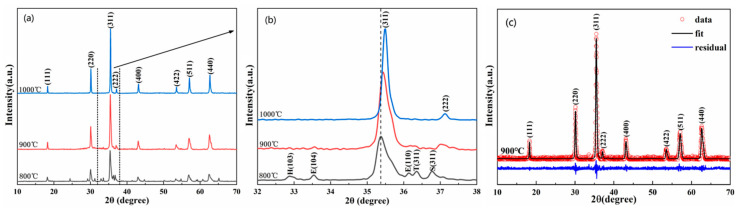
(**a**) X-ray diffraction (XRD) pattern of (FeCoCrMnZn)_3_O_4_ powders calcined at 800, 900, and 1000 °C, respectively; (**b**) partially enlarged XRD patterns (32–38°), where peaks indexed with (E), (H), (T), and (S) correspond to eskolaite, hetaerolite, cobalt manganese oxide, and Co_3_O_4_-like spinel phases, respectively; (**c**) Rietveld refinement XRD pattern of the powders calcined at 900 °C.

**Figure 2 materials-13-05798-f002:**
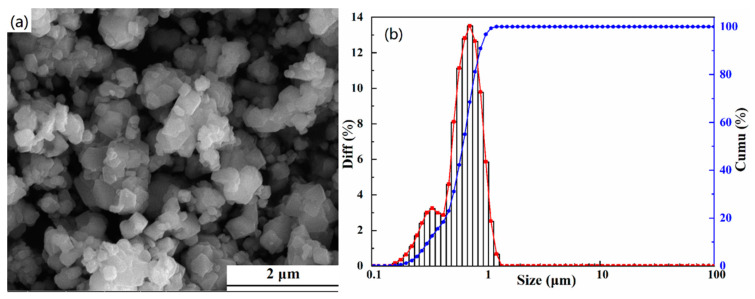
SEM image (**a**) and particle size distribution (**b**) of (FeCoCrMnZn)_3_O_4_ powders (900 °C, 2 h).

**Figure 3 materials-13-05798-f003:**
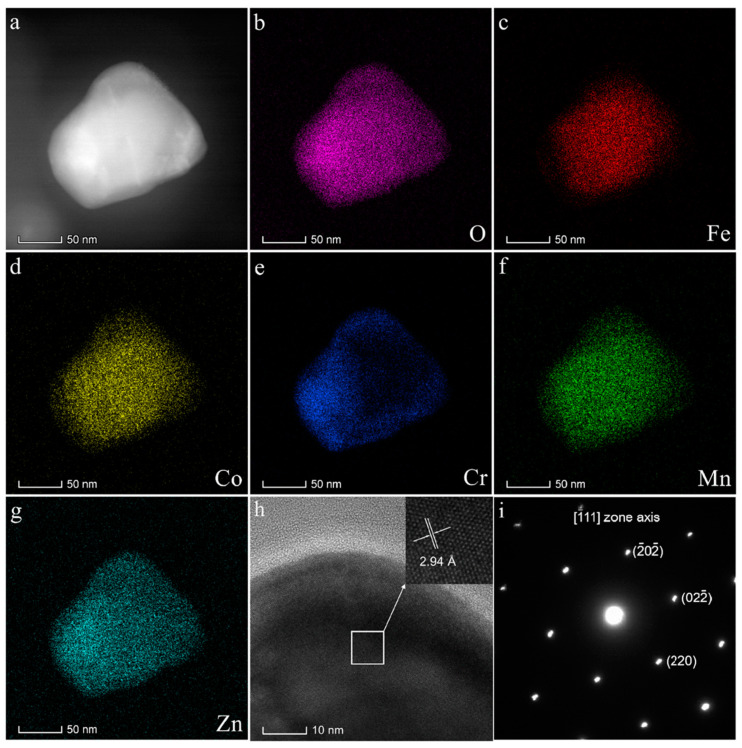
Characterization of (FeCoCrMnZn)_3_O_4_ powder: (**a**) STEM image, (**b**–**g**) energy-dispersive X-ray spectroscopy (EDS) analysis of O, Fe, Co, Cr, Mn, Zn, (**h**) high-resolution transmission electron microscope (HR-TEM) image, and (**i**) corresponding fast Fourier transform (FFT) pattern aligned along the $\left[\bar{1}11\right]$ zone axis.

**Figure 4 materials-13-05798-f004:**
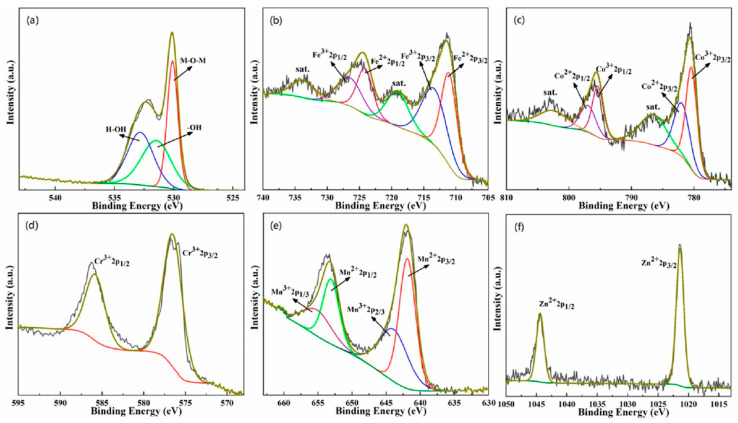
X-ray photoelectron spectroscopy (XPS) patterns of (FeCoCrMnZn)_3_O_4_ powders: (**a**) O 1*s*, (**b**) Fe 2*p*, (**c**) Co 2*p*, (**d**) Cr 2*p*, (**e**) Mn 2*p*, and (**f**) Zn 2*p*.

**Figure 5 materials-13-05798-f005:**
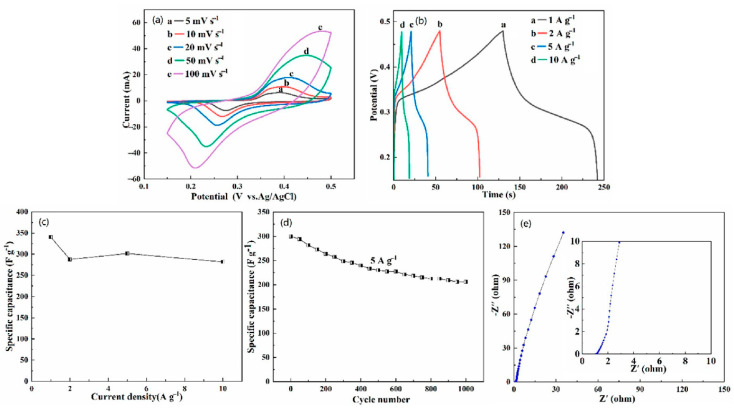
Electrochemical characteristics of the (FeCoCrMnZn)_3_O_4_/nickel foam (NF) composites: (**a**) cyclic voltammetry (CV) curves, (**b**) galvanostatic charge-discharge (GCD) curves, (**c**) corresponding specific capacitance (SCs) at different current densities, (**d**) cycling performance at a current density of 5 A·g^−1^, and (**e**) electrochemical impedance spectra (EIS).

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
