# Peer review of "Spinel-Type (FeCoCrMnZn)3O4 High-Entropy Oxide: Facile Preparation and Supercapacitor Performance"

_materials, 2020, doi:10.3390/ma13245798_

Round 1

Reviewer 1 Report

Dear Editor,

The authors presented their study on structural and electrochemical properties of (FeCoCrMnZn)3O4. I have some questions and remarks.

  1. What were the heating and cooling rate for (FeCoCrMnZn)3O4 powder synthesis?
  2. Why did you choose to study the properties of the powder obtained at 900ºC not that at 1000 ºC?
  3. What is the pore size of Ni-foam? How this affects the penetration of ferrite in?
  4. Figure 1c caption: I suggest to be mention of the synthesis temperature.
  5. Please, add the magnification bar on Figure 1a.
  6. I suppose that the Figure 2b presented the results of the particle size distribution obtained from laser particle size analyzer. As (FeCoCrMnZn)3O4 powder is magnetic, how the aggregation of particles due to magnetic force was influence on the particle size data?
  7. Please, improve the quality of Figure 5e.

Author Response

Ref. No.: Materials-1030568Title: Spinel Type (FeCoCrMnZn)3O4 High Entropy Oxide: Facile Preparation and Supercapacitor Performance

Thanks to the reviewers for your time and thoughtful comments, which have been incorporated into the revised manuscript. Hopefully we have addressed all of your concerns.

Our responses to the Reviewer’s comments are presented in BOLD font as follows. The page and line numbers refer to our revised manuscript submitted at 12/15/2020.

Reviewer #1: The authors presented their study on structural and electrochemical properties of (FeCoCrMnZn)3O4. I have some questions and remarks.

  1. What were the heating and cooling rate for (FeCoCrMnZn)3O4 powder synthesis?  Response: The heating rate is 10 °C/min and the cooling rate is about 2 °C/min (cooling with furnace) (in RED font, Line 60~63, Page 2). 
  2. Why did you choose to study the properties of the powder obtained at 900 °C not that at 1000 °C?  Response: We choose to study the properties of the powder obtained at 900 °C in consideration of following reasons: (i) Both (FeCoCrMnZn)3O4 powders obtained at 900 °C and 1000 °C possess single spinel structure. (ii) The (FeCoCrMnZn)3O4 powder obtained at 900 °C is smaller than that obtained at 1000 °C. (iii) Generally, smaller powder size (higher specific surface area) indicates better supercapacitor performance. 
  3. What is the pore size of Ni-foam? How this affects the penetration of ferrite?Response: As shown in the following SEM image of Ni-foam, the pore size is 100~300 mm. We suppose that there is no notable effect on the penetration of ferrite because the pore size is sufficiently large.SEM image of Ni-foam
  4. Figure 1c caption: I suggest to be mention of the synthesis temperature.  Response: The synthesis temperature was marked in Figure 1(c) caption and the title of the Figure 1 as suggested by the reviewer (in RED font, Line 114, Page 3).  
  5. Please, add the magnification bar on Figure 1a.  Response: The magnification bar was added into Figure 1(a) (Line 110, Page 3).
  6. I suppose that the Figure 2b presented the results of the particle size distribution obtained from laser particle size analyzer. As (FeCoCrMnZn)3O4 powder is magnetic, how the aggregation of particles due to magnetic force was influence on the particle size data?  Response: As the reviewer mentioned, (FeCoCrMnZn)3O4 powder is magnetic. In this work, to reduce agglomeration, the particle size of the powders was tested after ultrasonic dispersion for 30 min.
  7. Please, improve the quality of Figure 5e.  Response: Figure 5(e) was improved in accordance with the reviewer’s comment and the Figure 5 was updated in the revised manuscript and submission system (Line 184, Page 7).

Reviewer 2 Report

Our critical comments and questions are as follows:

  1. Line 61 - the authors indicate the temperature range of 800-1000C for annealing the spinel material. What was the optimal T and why? What was the atmosphere of annealing?
  2. What were the impurities in MnO2 of only 90% purity? How they ca impact the electrochemical performance of supercapacitors? The sources of the oxides used should be indicated, as well as the type of the ball-milling machine and the balls used.
  3. The authors should provide some characteristics of the NF. Is it electrochemically stable in the KOH solution?
  4. Lines 139-140 - What was the Mn valence state as estimated from XPS?
  5. Line 168 - The term "..is a highly diffused process" should be clarified.
  6. What instrument was used for the impedance studies?
  7. Line 191 - the authors' statement "...provides a new method for preparing HEO" should be confirmed by discussing its real novelty and advantages over the existing methods.
  8. Scale in Figure 2a should be indicated. 
  9. Some English grammar should be corrected. For instance: line 123 -...are highly; line 157 - ...was studied; line 80 - was measured.

Author Response

Ref. No.: Materials-1030568Title: Spinel Type (FeCoCrMnZn)3O4 High Entropy Oxide: Facile Preparation and Supercapacitor Performance

Thanks to the reviewers for their time and thoughtful comments, which have been incorporated into the revised manuscript. Hopefully we have addressed all of your concerns.

Our responses to the Reviewer’s comments are presented in BOLD font as follows. The page and line numbers refer to our revised manuscript submitted at 12/15/2020.

Reviewer #2: Our critical comments and questions are as follows: 

  1. Line 61 - the authors indicate the temperature range of 800°C-1000°C for annealing the spinel material. What was the optimal T and why? What was the atmosphere of annealing?  Response:In this work, (FeCoCrMnZn)3O4 powders was calcined at 800°C~1000°C in atmosphere (in RED font, Line 62, Page2). In this work, 900 °C should be the optimal temperature because: (i) Second phase existed in (FeCoCrMnZn)3O4 powders obtained at 800 °C; (ii) Both (FeCoCrMnZn)3O4 powders obtained at 900 °C and 1000 °C possess single spinel structure; (iii) The (FeCoCrMnZn)3O4 powder obtained at 900 °C is smaller than that obtained at 1000 °C. (iv) Generally, smaller powder size (higher specific surface area) indicates better supercapacitor performance. 
  2. What were the impurities in MnO2 of only 91.0% purity? How they impact the electrochemical performance of supercapacitors? The sources of the oxides used should be indicated, as well as the type of the ball-milling machine and the balls used.  Response:The oxides were purchased from Sinopharm Chemical Reagent Co., Ltd (in RED font, Line 60, Page 2). As shown in the following label images of MnO2, the information of impurities is not available, so the influence of the impurities in MnO2 on the electrochemical performance of supercapacitors is indeterminate. We will use MnO2 with higher purity in our future work to avoid the influence of impurities as possible as we can. The type of the ball-milling machine is GMJ-5 (Xianyang Jinhong General Machinery Ltd.) and the mill medium is ZrO2 ball. (in RED font, Line 61, 62, Page 2) Label images of MnO2
  3. The authors should provide some characteristics of the NF. Is it electrochemically stable in the KOH solution?  Response: As shown in the following SEM image of Ni-foam, the pore size is 100 mm~300 mm. To evaluate the electrochemical stability of Ni-foam in the KOH solution, CV test of Ni-foam in KOH solution had been carried out for 60 min and no detectable mass change of Ni-foam was found, before and after the test. Therefore, Ni-foam is electrochemically stable in the KOH solution.SEM image of Ni-foam
  4. Lines 139-140 - What was the Mn valence state as estimated from XPS?Response: Mn cations in (FeCoCrMnZn)3O4 HEO are at the valence of both +2 and +3 state (Line 142, 143, Page 5).
  5. Line 168 - The term "..is a highly diffused process" should be clarified.Response: We corrected it as “is a diffusion process” (in RED font, Line 171, Page 6)
  6. What instrument was used for the impedance studies?  Response: The supercapacitor performance (CV curves, GCD curves and EIS spectra) were measured by an electrochemical workstation (CHI660E, Shanghai Chenhua Instrument Co., LTD, China) (in RED font, Line 81, 82, Page 2)
  7. Line 191 - the authors' statement "...provides a new method for preparing HEO" should be confirmed by discussing its real novelty and advantages over the existing methods.  Response: Our former description is not appropriate, so we corrected it as “provides a facile method for preparing HEO” (in RED font, Line 194, Page 7).
  8. Scale in Figure 2a should be indicated.  Response: The scale bar was added into Figure 2(a) (Line 121, Page 4).
  9. Some English grammar should be corrected. For instance: line 123 -...are highly; line 157 - ...was studied; line 80 - was measured.  Response: We checked and corrected the grammatical mistakes (in BLUE font).

This manuscript is a resubmission of an earlier submission. The following is a list of the peer review reports and author responses from that submission.